# Bridging solvent molecules mediate RNase A – Ligand binding

**Stefan M. Ivanov** *, Ivan Dimitrov, Irini A. Doytchinova

Faculty of Pharmacy, Medical University of Sofia, Sofia, Bulgaria

* sivanov@ddg-pharmfac.net

**Data Availability Statement:** All relevant data are within the manuscript and attached figures and S1 File.

**Funding:** SMI, ID, and IAD were awarded funding via Grant No. D01-221/03.12.2018 for NCHDC – part of the Bulgarian National Roadmap on RIs.

## Abstract

Due to its high catalytic activity and readily available supply, ribonuclease A (RNase A) has become a pivotal enzyme in the history of protein science. Moreover, this great interest has carried over to computational chemistry and molecular dynamics, where RNase A has become a model system for various types of studies, all the while being an important drug design target in its own right. Here, we present a detailed molecular dynamics study of RNase–ligand binding involving 22 compounds, spanning nearly five orders of magnitude in affinity, and totaling 8.8 μs of sampling with the standard Amber parameters and an additional 8.8 μs of sampling with a modified potential. We show that short-lived, solvent-mediated bridging interactions are crucial to RNase–ligand binding. We characterize the behavior of bridging solvent molecules, uncovering a power-law dependence between the lifetime of a solvent bridge and the probability of its occurrence. We also demonstrate that from an energetic perspective, bridging solvent in RNase A–ligand binding behaves like part of the enzyme, rather than the ligands. Moreover, we describe an automated pipeline for the detection and processing of bridging interactions, and offer an independent assessment of the performance of the state-of-the-art fixed-charge force fields. Thus, our work has broad implications for drug design and computational chemistry in general.

## Introduction

Ribonuclease A (RNase A) is a member of the endonuclease family of enzymes that degrade RNA. It and its homologs are involved in a multitude of functions in both the healthy and diseased state [1,2]. Some of the more prominent of these homologs are the eosinophil-derived neurotoxin (RNase 2), and eosinophil cationic protein (RNase 3), which have been implicated in hypereosinophilic and allergic conditions, and angiogenin (RNase 5), which induces neovascularization during normal organ growth, cancer, and in vascular and rheumatoid disorders [3]. Furthermore, RNase A has been shown to have the highest catalytic activity of all known family members. In part due to its abundance and high catalytic activity, bovine pancreatic ribonuclease became the first enzyme to have its catalytic mechanism described [4]. It is, therefore, unsurprising that bovine pancreatic ribonuclease A has been the subject of many pioneering studies in protein chemistry and enzymology, and is commonly used in laboratories. Moreover, apart from experimental studies, it has also served as a model system for

The funders played no role in the study design, data collection and analysis, decision to publish, or preparation of the manuscript.

**Competing interests:** The authors have declared that no competing interests exist.

molecular dynamics investigations focusing on enthalpy-entropy compensation, ligand binding entropy, protein engineering for thermal stability, and the role of protein dynamics in enzymatic catalysis [5]. Finally, RNase A and its homologs have become important targets for drug design [6,7]. Previous crystallographic studies [8] have suggested bridging water molecules [9,10] are involved in RNase–ligand binding, possibly forming a network of water-mediated interactions [11]. The extent of solvent involvement in RNase—ligand binding, however, remains uncertain.

Here, we present a molecular dynamics investigation of the binding between full-length bovine pancreatic RNase A and 22 nucleotide, nucleoside, and pyrophosphate-containing ligands, spanning nearly five orders of magnitude in affinity (Fig 1; note that in the figure phosphate groups are *drawn* protonated, whereas they have been *simulated* in a fully deprotonated state). We show that bridging water molecules and ions are an essential component of RNase A–ligand binding, and characterize this interaction at resolution and in detail that is unavailable to structural studies. We demonstrate that a far greater number of water molecules mediate the binding process than what is visible in published RNase–ligand structures [6–15], and characterize their behavior and contribution to the binding process. Our work involves extensive sampling–nearly 10 μs with the standard Amber parameters and nearly 10 μs with a modified potential. This serves as an independent test of the latest version of the general Amber force field–GAFF 2.11 [16]. Moreover, our work has broad implications for drug design campaigns targeting RNases, as well as for the broader field of computer-aided drug design (CADD) [17] in general.

## Methods

### System preparation

Initial coordinates for the complexes between RNase A and 22 different ligands (Fig 2) were obtained from the 2018 refined set of the PDBbind database [18,19]. Water molecules up to 4 Å away from ligand atoms were retained to ensure that all waters mediating protein–ligand interactions from the published structures are included in the simulations. As all 22 structures included the full-length protein with no chain breaks, caps were not needed. Protein chains were protonated and solvated in a truncated octahedral box with TIP3P water [20] with the *tleap* program from the Amber18 package with a minimal wall distance of 15 Å, for a total system size of around 30 000 atoms. 0.150 M of NaCl with Joung and Cheatham parameters [21] were added to approximate a physiological salt concentration while ensuring charge neutrality. Ligand protonation states and net charges were set to those distributed by the PDBbind curators; only for ligands with an odd number of bonding electrons were charges manually adjusted to their correct values, corresponding to a pH of 7. Ligand parameters were obtained using the general Amber force field (GAFF 2.11) [16] with AM1-BCC charges [22] using *antechamber*.

### Molecular dynamics simulations

The solvated systems were subjected to a series of 1000 steps of energy minimization with the steepest descent method and harmonic restraints of 3 kcal*mol$^{-1}$*Å$^{-2}$ applied to all heavy atoms of the solutes (ligand and protein), followed by 1000 steps of conjugate gradient minimization with identical restraints. The systems were heated from 0 to 300 K over a period of 1 ns at constant volume with 3 kcal*mol$^{-1}$*Å$^{-2}$ harmonic restraints on solute heavy atoms, followed by 1 ns of constant pressure density equilibration with restraints. The systems were then equilibrated for 1 ns without any restraints and simulated for 100 ns of production dynamics under constant pressure (1 bar) and temperature (300 K) conditions, maintained with the

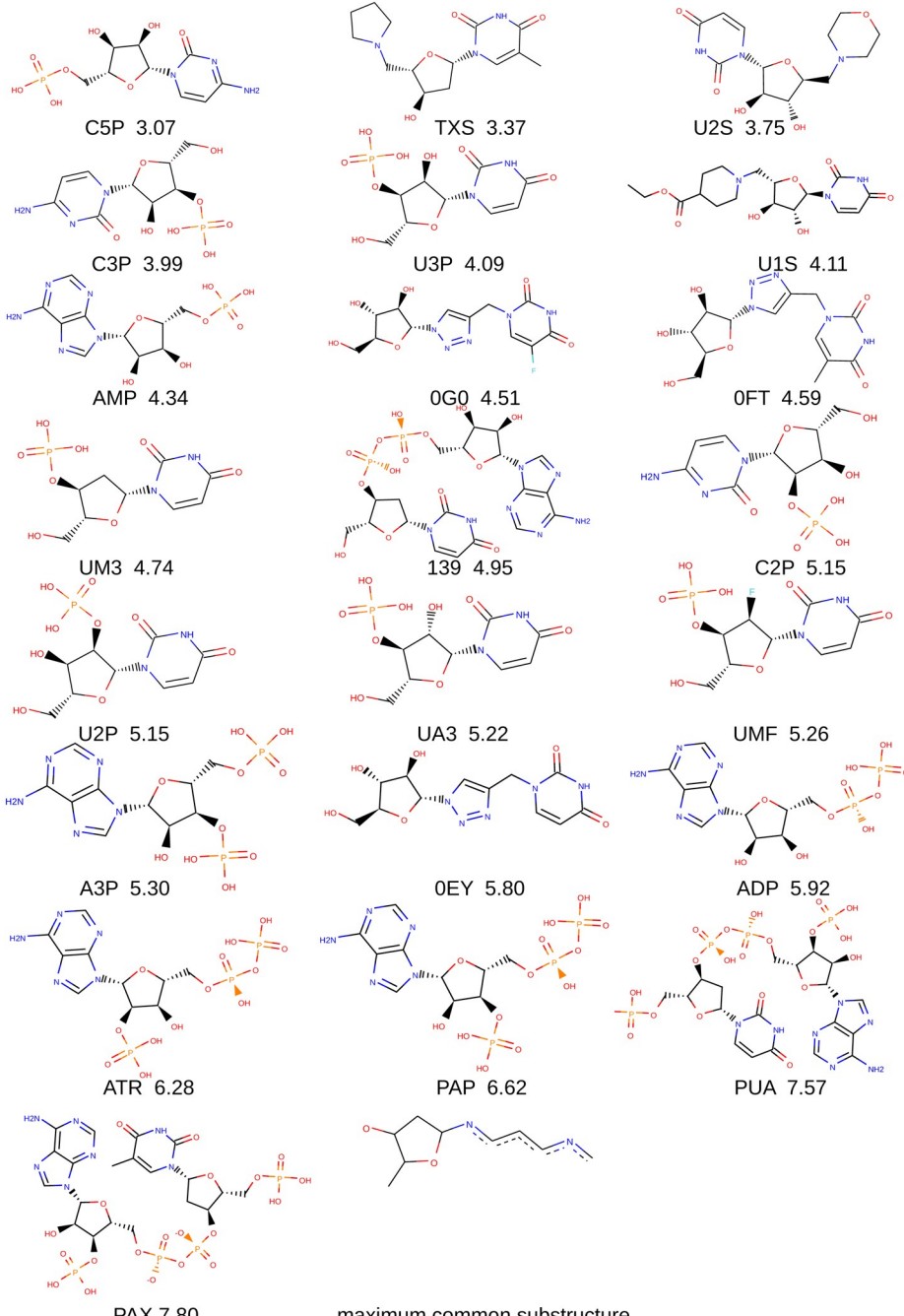

**Fig 1. Ligand molecules examined in the present study and their maximum common substructure.** Below each ligand, we give its name and its p$K$ value for RNase A. All p$K$ values are p$K_i$s, except for PAX, which is a p$K_d$. Therefore, throughout the present report, we refer to these values simply as p$K$s. Note that phosphate groups are drawn protonated, whereas they have been simulated in a fully deprotonated state.

Berendsen barostat [23] and the Langevin thermostat, respectively [24]. Collision frequencies for temperature coupling were set to 2 ps$^{-1}$; the pressure relaxation time was set to 2 ps, using isotropic position scaling. All systems were simulated in four independent replicas with the ff14SB force field under periodic boundary conditions [25]. A 12.0 Å cutoff was used for both

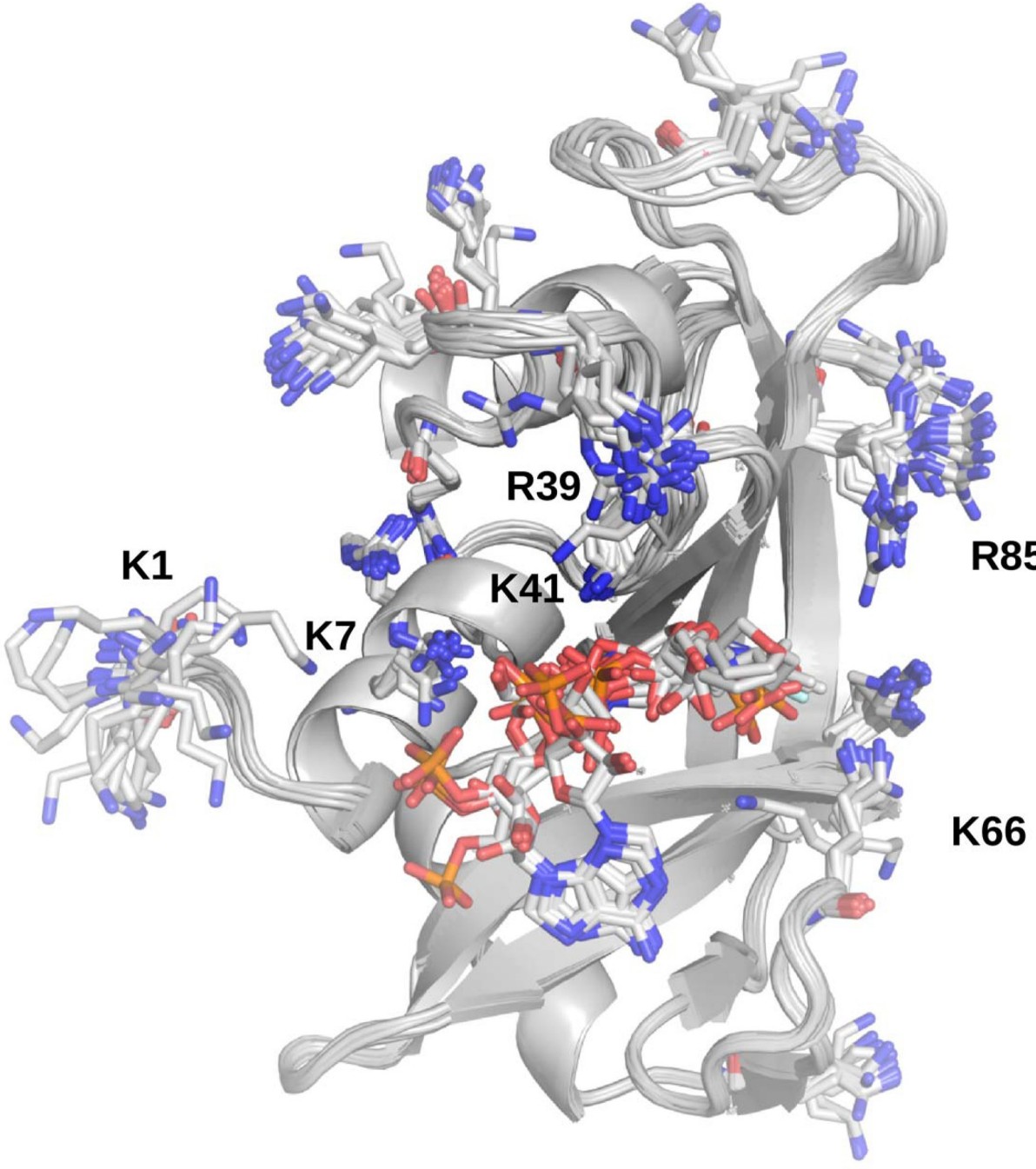

**Fig 2. The 22 starting protein–ligand crystal and NMR structures aligned by Cα.** The protein backbone is in cartoon representation, colored in gray. Lysines, arginines, and ligands are given in stick representation with carbon atoms in white, nitrogens in blue, oxygens in red, and phosphorus in orange. The view is centered on the ligands in the binding groove of the enzyme. Also labeled are several lysine and arginine residues, which are further discussed in the text.

van der Waals and electrostatic interactions; long-range electrostatics beyond the real-space cutoff were computed with the particle-mesh Ewald (PME) scheme [26]. During heating, density equilibration, preproduction, and production dynamics, bonds to hydrogen were constrained using the SHAKE algorithm [27], allowing for a 2 fs time step; only during energy

minimization bonds to hydrogen were not constrained. During production dynamics, frames were saved every 100 ps (0.1 ns) for a total of 1000 per trajectory, to be used in subsequent analysis.

## Trajectory processing, identification of bridging solvent, and residence time analysis

Rototranslational alignment, as well as ligand heavy atom and protein carbon alfa (Cα) root-mean-square deviation (RMSD), and root-mean-square fluctuation (RMSF) calculations (with respect to the starting coordinates) were performed with *cpptraj V4.14.0* [28]. For each production trajectory, bridging solvent molecules and ions were identified with the *hbond* command for *cpptraj* using the *nointramol* keyword, which excludes intramolecular bridging waters and ions from consideration. The distance and angle cutoffs for *hbond* were 3 Å and 135 degrees, respectively. After all intermolecular bridging agents were identified, they, along with the protein and ligand, were stripped into a separate trajectory file. Corresponding topologies for these trajectories were generated with the *ante-MMPBSA.py* utility of AmberTools19.

For each production dynamics run, the number of bridging solvent molecules and ions was calculated, as were the total number of bridging interactions formed during the 100 ns simulations, and the residence times of the solvent molecules. Herein, we define a bridging water molecule as one that is simultaneously hydrogen bonded to the protein and ligand in at least one of the 1000 frames per production dynamics run. When counting the number of bridges per trajectory, we defined each bridge as a single, continuous interaction. For example, if a given water molecule is involved in bridging interactions in frames 300, 390, 391, 392, 911, and 912, we record three separate bridges with lifetimes of 0.1, 0.3, and 0.2 ns, respectively. We stress that a given water molecule or ion can be involved in multiple bridging interactions. Hence, the number of bridges in a molecular dynamics simulation is typically much larger than the number of bridging molecules. For example, an ion may become involved in a single, long-residence time bridging interaction where it remains located between the ligand and protein for several nanoseconds without interruption. Then, it can leave this location and return to it multiple times for short periods of time. Hence, we record one bridging ion and several bridging interactions–one long-lived and multiple short-lived bridges. Due to the ambiguity in defining "residence time" throughout the literature, we point out that in the present report, we use the terms "lifetime of a bridging interaction" and "residence time of a water molecule/ion" interchangeably.

To gain further insight into the global behavior of the bridging waters and ions over the course of the 400 ns of production dynamics for each complex, we generated their radial distribution functions (RDFs) around the ligands with the Gromacs *rdf* tool. We have performed RDF analysis for bridging water and bridging ions separately.

## MM-PB(GB)SA calculations and analysis

For each complex trajectory, the enthalpy of interaction between the protein and the bound ligand (ΔH) was computed with the MMPBSA.py script [29], part of the Amber18 package. Briefly, molecular mechanics—Poisson-Boltzmann surface area (MM-PBSA) calculations are an implicit solvent, end-state free energy method [30]. A thermodynamic cycle is constructed and the energy of interaction is evaluated as the difference between the complex free energy and the sum of the individual components' free energies:

$$\Delta G_{binding} = \langle \Delta G_{complex,solvated} \rangle - (\langle \Delta G_{protein,solvated} \rangle + \langle \Delta G_{ligand,solvated} \rangle) \tag{1}$$

where the angle brackets denote an ensemble average of snapshots taken from a (usually

explicit solvent) trajectory [31]. For each component, ΔG is evaluated as

$$\Delta G = \langle E_{gas} \rangle + (\langle \Delta G_{solvation} \rangle - T \langle S_{solute} \rangle) \tag{2}$$

where $E_{gas}$ is a gas phase energy, calculated from the trajectory in accordance with the force field parameters, $\Delta G_{solvation}$ is the solvation free energy, computed with an implicit solvent model, and TS is the entropic contribution to binding, estimated via normal mode analysis or with the quasiharmonic approximation [29,32]. The solvation energy, in turn, comprises an electrostatic and nonpolar component:

$$\Delta G_{solvation} = \Delta G_{electrostatic} + \Delta G_{nonpolar}$$

The former is computed by modeling the solute as a set of spheres with appropriate charges and radii, embedded in a structureless continuum (the solvent and ions dissolved in it) and numerically solving the Poisson-Boltzmann equation [33]. Poisson-Boltzmann calculations were performed using the internal PBSA solver in *sander*. The nonpolar part was computed as a sum of two terms–a repulsive term stemming from the formation of a cavity in the solvent, which accommodates the solute, as well as from repulsive solute–solvent interactions, and an attractive term, arising from favorable solute–solvent interactions [34].

$$\Delta G_{nonpolar} = \Delta G_{repulsive} + \Delta G_{attractive}$$

It has been shown that the attractive term can be approximated by the van der Waals interactions between solute and solvent [35,36] which are modeled with the Lennard-Johnes 6–12 potential. Finally, the repulsive component of the nonpolar free energy of solvation is obtained via a simple linear relationship with the solvent accessible surface area (SASA) of the solute:

$$\Delta G_{repulsive} = \gamma * SASA + c$$

where $\gamma$ is a surface tension coefficient and c is a cavity offset corresponding to $\Delta G_{repulsive}$ for a solute of zero volume [34].

In our subsequent analysis, we also included the similar in spirit but computationally cheaper molecular mechanics—generalized Born surface area calculations (MM-GBSA). Like MM-PBSA, MM-GBSA is also an end state, post-processing method for evaluating binding free energies. Here, polar solvation energies are computed from pairwise summations over charge–charge interactions, scaled in accord with effective atomic burial or the "Born radius" of the atoms [37]. Born radii were obtained from the work of Onufriev et al. [38] by setting the *igb* parameter to 5. SASA is calculated with the linear combinations of pairwise overlaps (LCPO) method from atomic radii [39].

MM-PBSA and MM-GBSA calculations were performed using *mbondi2* radii and default settings for the nonpolar decomposition scheme, surface tension, cavity offset, and external and internal dielectric constants. We only adjusted the default setting for the ionic strength (0.0 M) to the one used during production dynamics– 0.150 M.

MM-PBSA and MM-GBSA calculations were performed on the complex trajectories (the so-called "one trajectory approach") on the complexes between the protein and the ligands, as well as complexes including the protein, ligand, and all bridging waters and ions.Per-residue energy decompositions were also performed for every individual frame, adding 1–4 energy terms to internal energy terms; the energies for every residue were averaged over the 1000 frames of production dynamics. This allowed us to assess each individual residue's contribution to binding over the course of an entire trajectory–favorable, unfavorable, or indifferent.

As we were primarily interested in relative binding energies, rather than their absolute values, and bearing in mind the considerable approximations and inaccuracies involved in

computing entropy [30,40,41], as well the considerable time and computational effort required, we chose to omit entropy calculations from our analysis. Given that entropies were not explicitly calculated, i.e. the entropy term has not been included in Eq (2), our MM-PB (GB)SA calculations produced the enthalpy of binding ($\Delta H$), which can be calculated rather reliably, as opposed to the free energy ($\Delta G$) and entropy ($\Delta S$), which cannot.

## Results

### Complex stability

The protein components of the complexes remained stable throughout the 100 ns dynamics runs in all replicas. The mean Cα RMSDs typically varied below 2.5 Å, with only individual trajectories exceeding 3 Å (see S1 Fig and S1 File). A more detailed analysis of the proteins was performed by examining the individual Cα fluctuations (S2 Fig and S1 File). This analysis revealed that the proteins were highly stable, with only unstructured regions, primarily the N-terminal loop, exhibiting high fluctuations (see Figs 2 and S2 and S1 File). The ligands exhibited somewhat varied behavior–most were stably bound, as assessed by their heavy atom RMSDs, although some exhibited greater mobility. This pertains to some of the smaller compounds, which tended to explore more of the relatively broad and shallow binding groove of the RNase (see, for example, Fig 3A and 3B), as well as the large, pyrophosphate (also known as diphosphate)-containing compounds, which reorient themselves to make extensive phosphate–lysine/arginine and pyrophosphate–lysine/arginine interactions with the protein (Fig 3C and 3D). As lysine and arginine side chains bear terminal amino groups, these interactions are of the amine–phosphate type. Most (pyro)phosphate–amine interactions involved K7, R10, R39, and K41 (Fig 3C and 3D), which form the binding groove of the enzyme. During dynamics, the amine–phosphate contacts often form clusters of salt bridges and salt-linked triads, where two salt bridges share a residue in between [42,43] (see the salt-linked triad featuring K7 in Fig 3D). Such a rearrangement is also observed with the small, phosphate-containing compounds where the crystal structures have little or no direct amine–phosphate contacts (see Fig 4A and 4C). After 100 ns of explicit solvent dynamics, the compounds reorient themselves to make extensive amine–phosphate contacts. In the C5P simulations, for example, the ligand becomes involved in salt-linked triads with R39 and K41 (compare Fig 4A and 4B). An even greater clustering occurs in the A3P simulations, as this compound has two phosphate groups, and forms a cluster of amine-phosphate contacts with K7, K10, K47, and K41 (compare Fig 4C and 4D).

### Solvent-mediated bridging interactions

Trajectory processing revealed that in each protein–ligand system, hundreds to thousands of different solvent molecules participated in at least one bridging interaction (Fig 5A). In the majority of trajectories, this corresponded to bridging interactions numbering in the thousands. Phosphate-containing compounds tended to attract more bridging waters than compounds that did not have phosphate groups. Furthermore, the larger, pyrophosphate-containing compounds tended to attract more bridging waters than the small compounds that have only one phosphate group (see Figs 1 and 5A). Moreover, examining the composition of the bridging agents showed that no chloride ions are involved in bridging interactions, i.e. only water and sodium ions mediate RNase–ligand binding. We then examined the duration of these bridging interactions. It was found that the vast majority of bridges (98 to over 99%) have rather short lifetimes–below 1 ns. However, among the different RNase–ligand systems, we also observed tens of bridges with lifetimes between 1 and 10 ns, and a small number of bridges, single-digit numbers for most compounds, that lasted 10 or more nanoseconds. The three longest-lasting

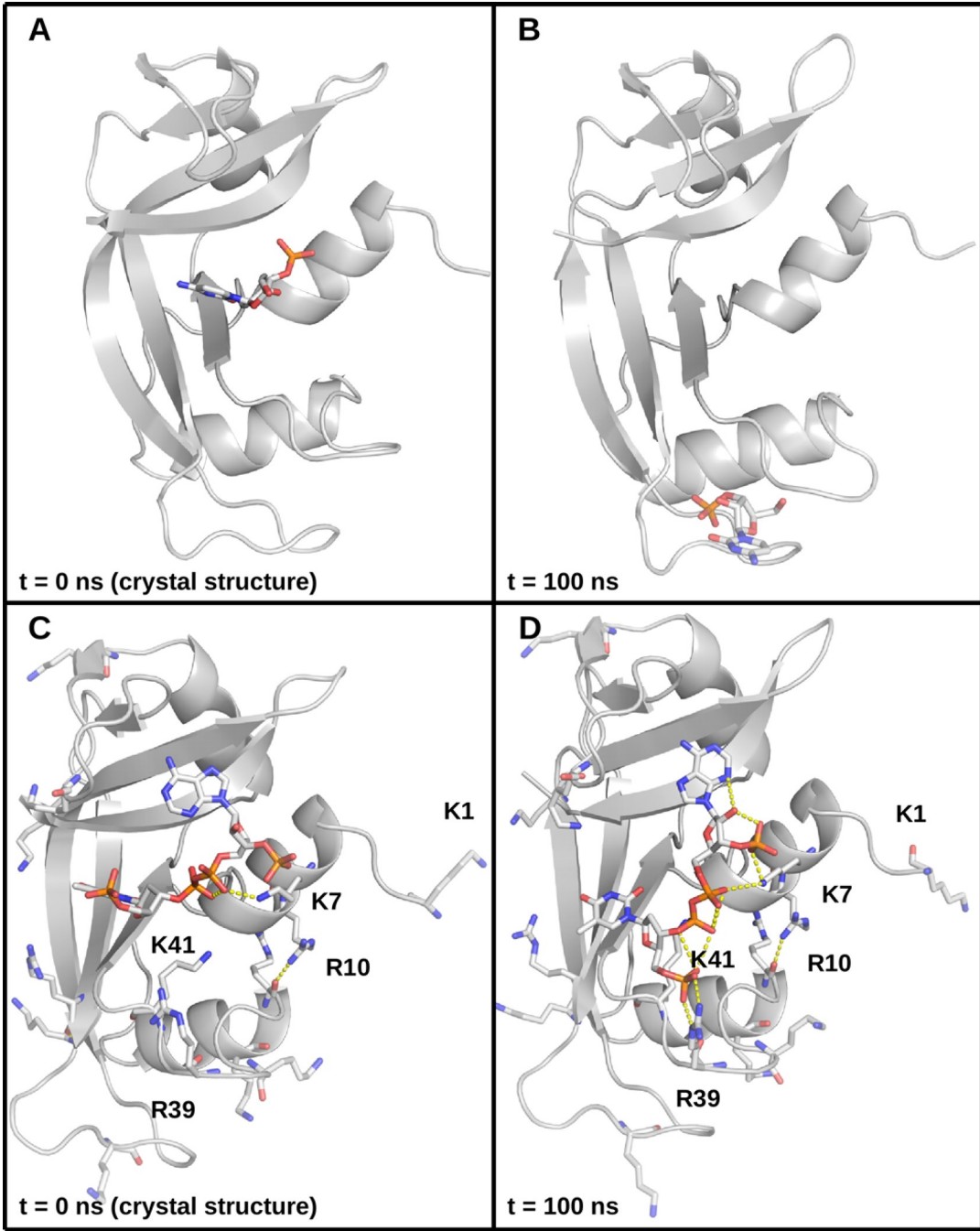

**Fig 3. Comparison between crystal and simulated structures.** (A) The RNase A–C3P complex from a crystal structure. The protein backbone is in cartoon representation, colored in gray, the ligand is given in stick representation with carbon atoms in white, nitrogens in blue, oxygens in red, and phosphorus in orange. (B) The RNase A–C3P complex after 100 ns of production dynamics. (C) The RNase A–PAX complex from a crystal structure. The protein backbone is in cartoon representation, colored in gray. Lysines, arginines, and the ligand are given in stick representation with carbon atoms in white, nitrogens in blue, oxygens in red, and phosphorus in orange. Certain lysine and arginine residues are labeled; polar contacts are shown as dotted lines. (D) The RNase A–PAX complex after 100 ns of production dynamics.

water bridges in our data set were observed in the PUA–RNase, C3P –RNase, and PAX–RNase systems, and had lifetimes of 55.5, 35.6 and 23.3 ns, respectively.

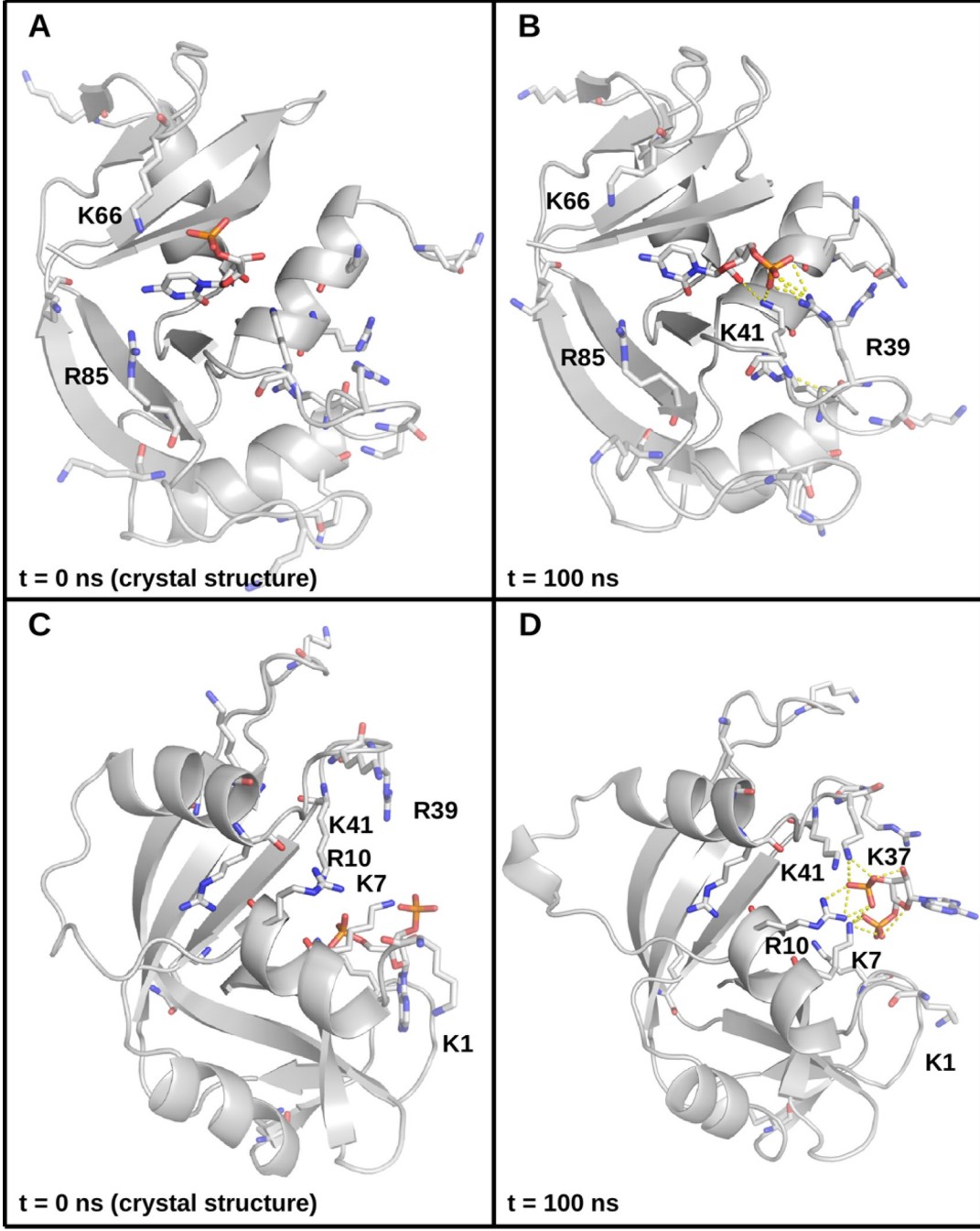

**Fig 4. Comparison between crystal and simulated structures.** (A) The RNase A–C5P complex from a crystal structure. The protein backbone is in cartoon representation, colored in gray. Lysines, arginines, and the ligand are given in stick representation with carbon atoms in white, nitrogens in blue, oxygens in red, and phosphorus in orange. Certain lysine and arginine residues are labeled. (B) The RNase A–C5P complex after 100 ns of production dynamics; polar contacts are shown as dotted lines. (C) The RNase A–A3P complex from a crystal structure. The protein backbone is in cartoon representation, colored in gray. Lysines, arginines, and the ligand are given in stick representation with carbon atoms in white, nitrogens in blue, oxygens in red, and phosphorus in orange. Certain lysine and arginine residues are labeled; polar contacts are shown as dotted lines. (D) The RNase A–A3P complex after 100 ns of production dynamics.

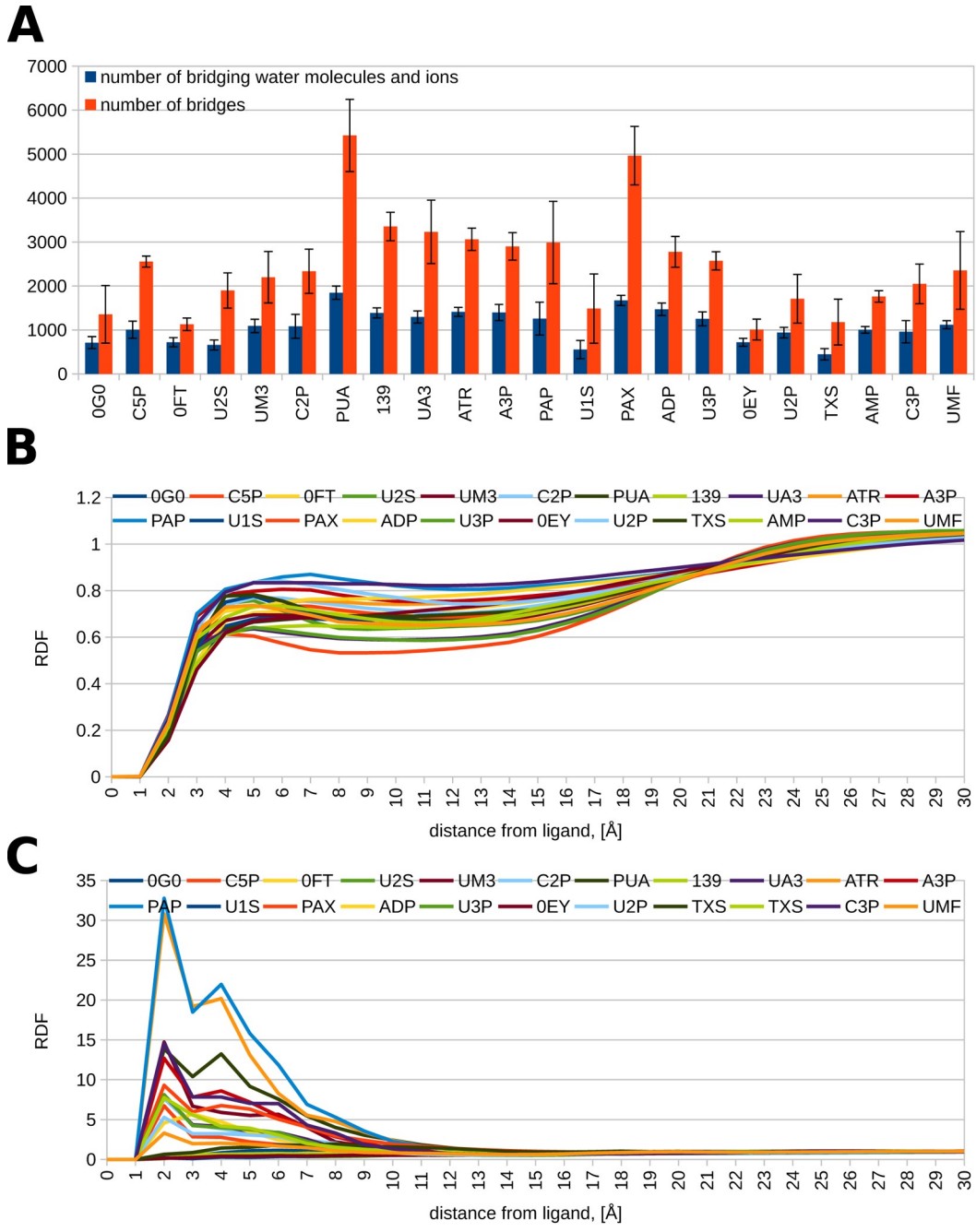

**Fig 5. Average numbers of bridging water molecules and ions, and bridging water–ligand and bridging Na$^+$—ligand RDFs.** (A) For every compound, we give the average number of bridging molecules (blue columns) and bridges (red columns) for the four production runs; error bars represent standard deviations. (B) Bridging water–ligand radial distribution functions computed from the four 100 ns production dynamics simulations for every compound. (C) Bridging Na$^+$–ligand radial distribution functions computed from the four 100 ns production dynamics simulations for every compound.

Furthermore, we examined the radial distribution functions of the bridging waters and bridging Na$^+$ around the ligands throughout the production dynamics runs. The radial distribution function provides information how likely it is for a molecule or chemical group to

be found at a given distance from another given group or molecule, as opposed to finding it in bulk solvent, where RDF = 1. In other words, RDF values below 1 indicate that a given molecule is less likely to be found around a given molecule at a given distance compared to being found in bulk solvent; RDF values above 1 indicate the opposite. Examining the RDFs for bridging waters and bridging $Na^+$ reveals an interesting pattern. It is evident that a bridging water molecule is more likely to be found in bulk solvent (Fig 5B and S1 File), rather than within 3 Å of the ligands, mediating their interactions with the protein. Conversely, for most ligands, sodium ions are more likely to be found in the vicinity of the compounds, rather than bulk solvent (Fig 5C and S1 File). Notably, the $Na^+$- ligand radial distribution functions for the pyrophosphate (or equivalently, diphosphate) compounds have high peaks, implying that bridging sodium ions are tens of times more likely to be found near the ligand, rather than in bulk solution (see Figs 1 and 5 and S1 File). Conversely, most of the compounds whose $Na^+$ RDFs do not exhibit peaks lack phosphate groups (see Figs 1 and 5 and S1 File).

## MM-PB(GB)SA calculations and analysis

We monitored the MM-PBSA and MM-GBSA energies over the course of the production dynamics runs to assess their convergence. When bridging solvent is excluded from the computations, most enthalpies converge (i.e., reach a plateau) well before the end of the simulation, around the 50 ns mark (S4 Fig and S1 File). Including bridging solvent in the calculations makes the systems larger and more complex. Correspondingly, most of the energies take longer to converge (S5 Fig and S1 File); this is particularly the case with the larger, pyrophosphate-containing compounds, e.g. PAP, PUA, and PAX. We note that a pyrophosphate group, known also as a diphosphate group, is simply two phosphate groups condensed together. These compounds tended to attract the greatest number of bridging solvent molecules, numbering in the thousands (see also Figs 1 and 5A). Nevertheless, there is no significant difference between results performed on the entire production runs and results derived only from the latter half of each trajectory (see the data in S1 File). Therefore, for completeness, from this point forward, we present and discuss only the former.

When excluding bridging particles from the MM-PBSA and MM-GBSA calculations and analysis, i.e. performing calculations on only the protein and the ligand, no correlation is observed between calculated ΔH values and experimental affinities, expressed as p$K$ (Fig 6 and S1 File). Moreover, the lines of best fit for the ΔH/p$K$ correlations from the four replicas cross each other at large angles. Conversely, when including bridging molecules in the calculations, the correlations become considerable, with the slopes of the lines of best fit from the four replicas becoming much more similar. A more detailed inspection of the ligands in Fig 1, the plots in Fig 6C and 6D, and the molecular dynamics trajectories reveals that the greatest outliers are phosphate or pyrophosphate containing molecules whose interactions with the protein are dominated by (pyro)phosphate–lysine or (pyro)phosphate–arginine interactions. These compounds have greatly overestimated affinities for the enzyme, noticeably eroding $R^2$.

As PAX follows the same general ΔH/p$K$ trend as the other ligands, we conclude that its p$K_d$ value is directly comparable to the p$K_i$ values of the remaining compounds.

## Discussion

We chose to utilize the PDBbind database, as it contains a large, manually-curated compilation of high-quality protein–ligand structures. Moreover, the refined set contains only p$K_i$ and p$K_d$ values, which are directly comparable to each other, unlike p$IC_{50}$ values, which depend critically on experimental conditions and can only be compared across identical assays. The

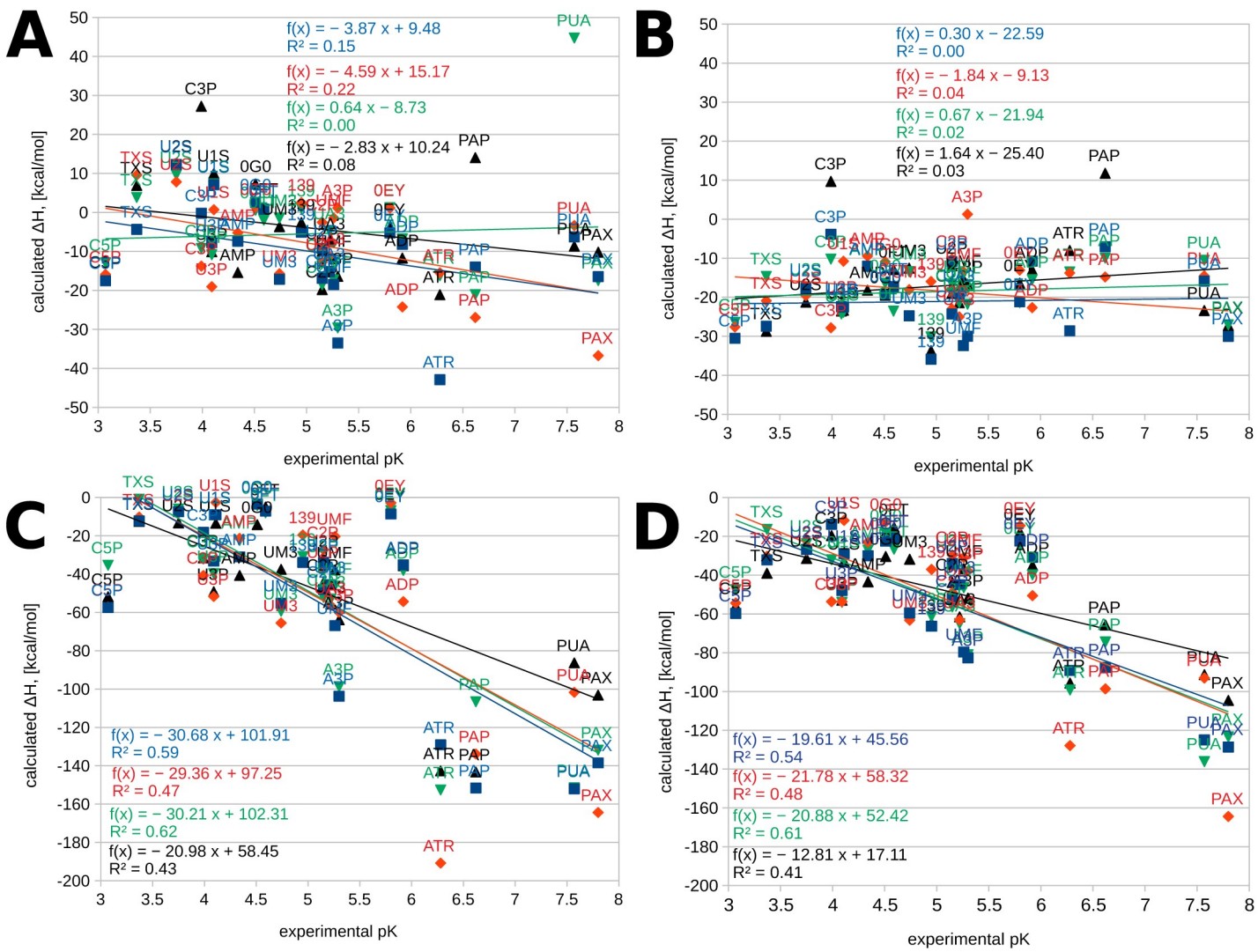

**Fig 6. ΔH/p*K* correlations.** (A) Correlations between computed enthalpy (ΔH) and experimental affinity from the MM-PBSA calculations from replica 1 (blue), replica 2 (red), replica 3 (green), and replica 4 (black); bridging waters and ions are not included in the calculations. For each replica, the line of best fit and the corresponding equation are also given in the respective color. (B) Correlations between computed enthalpy (ΔH) and experimental affinity from the MM-GBSA calculations from replica 1 (blue), replica 2 (red), replica 3 (green), and replica 4 (black); bridging waters and ions are not included in the calculations. For each replica, the line of best fit and the corresponding equation are also given in the respective color. (C) Correlations between computed enthalpy (ΔH) and experimental affinity from the MM-PBSA calculations from replica 1 (blue), replica 2 (red), replica 3 (green), and replica 4 (black); bridging waters and ions are included in the calculations. For each replica, the line of best fit and the corresponding equation are also given in the respective color. (D) Correlations between computed enthalpy (ΔH) and experimental affinity from the MM-GBSA calculations from replica 1 (blue), replica 2 (red), replica 3 (green), and replica 4 (black); bridging waters and ions are included in the calculations. For each replica, the line of best fit and the corresponding equation are also given in the respective color.

PDBbind database contains only molecules consisting of C, N, O, P, S, F, Cl, Br, I, and H atoms and excludes ligands with unusual chemistries, e.g. compounds containing Be, B, Si or metals. Thus, it is particularly well suited to the needs of drug design.

As in previous work [44,45], we decomposed $\Delta G_{nonpolar}$ into a dispersive (attractive) and cavitation (repulsive) term [34] in the MM-PBSA calculations, as this scheme was shown to provide much better agreement between computed [44] and isothermal calorimetry results [46] for ΔH. The alternative scheme, where $\Delta G_{nonpolar}$ is linearly dependent on SASA, has been shown to grossly overestimate ΔH [44].

The current implementation of MM-PB(GB)SA calculations in MMPBSA.py [29] requires that the receptor and ligand be explicitly defined. When including bridging waters and ions in the calculations, this creates an ambiguity–one can define these particles either as part of the receptor or the ligand. Previous work on a diverse range of systems, including topoisomerase–camptothecin derivatives, α-thrombin–benzothiophene and benzopiperidine derivatives, penicillopepsin–peptide, penicillopepsin–naphthalene derivative complexes, and avidin–biotin analogues [47], has shown that computed ΔH correlates with affinity to a significant degree only when the bridging solvent is treated as part of the receptor. Our results on the complexes between bovine pancreatic RNase and 22 nucleotides, nucleosides, and their analogs corroborate this finding. Indeed, the $R^2$ values presented in Fig 6C and 6D were obtained by including bridging water molecules and ions in the receptor; including them in the ligand produces no significant correlation (see the data in S1 File). It is, therefore, evident that this behavior is consistent across different systems, rather than being system-dependent. Moreover, this behavior is perhaps unexpected and certainly warrants rationalization. High-resolution structural studies on the oligopeptide binding protein (OppA) complexed to different tripeptides of the KXK type, where only the middle residue is allowed to vary, show that different side chains allow for a different water content in the binding cavity of the protein, but the waters that remain in common occupy the same positions from one peptide to another [48]. This has prompted some authors to suggest that in a sense, water acts as part of the protein, moving around to change the shape of the binding cavity by selecting from a predefined set of conserved positions [49]. Intriguingly, we arrive at a similar conclusion after approaching the problem from a different angle. Our independent study, therefore, complements previous structural work, adding an energetic perspective, and corroborates this hypothesis.

Our results on bridging waters are in semiquantitative agreement with published structural studies on ligand binding to bovine pancreatic RNase, where a small number (usually < 10) of bridging water molecules is observed, thereby lending credence to the TIP3P water model, the ff14SB, and general Amber force field 2.11, and their compatibility. It is also instructive to compare published structural data with the behavior of the complexes in our molecular dynamics simulations. For example, crystal structures of the PUA–RNase [7] and PAX–RNase [50] complexes show that the two compounds form bridging interactions with the protein on both ends, through their uracyl and adenosyl, and thymine and adenosyl groups, respectively (Fig 7A and 7D).We observed that during dynamics, these water molecules rapidly exchange with others, while generally maintaining similar networks of interactions to those from the crystal structures (compare the starting structure in Fig 7A, 7B and 7C). Moreover, in certain trajectories, the ligands tended to reorient themselves to make more amine–phosphate interactions with the protein, which was accompanied with reorganization of the bridging interactions (compare 7D to 7E and F, also see Fig 3C and 3D). Interestingly, sodium ions can also become involved in the bridging network formed by the adenosyl group (Fig 7C). Additionally, in the course of dynamics, sodium ions tended to approach the phosphates and form long-lived bridging interactions among the phosphates and negatively charged or polar side chains from the protein, along with backbone oxygen atoms (see, for example, Fig 7B, 7C, 7E, 7F, 7H and 7I). One such example is the RNase–C3P complex [8], where the ligand drifted from its starting position in the center of the groove to its edge, forming a long-lived interaction with a sodium ion and E86, along with nearby backbone and side chain oxygen atoms (see Figs 7G, 7H, 7I and 3A and 3B). Trajectory analysis, together with the data presented in Fig 5, paints a picture of numerous (hundreds to thousands) bridging water molecules, which rapidly exchange with each other, and much less numerous (usually below 10) sodium ions, which tend to form longer-lived bridging interactions. Indeed, the lack of any bridging Cl⁻ in our entire data set likely points to the importance of the phosphate groups for binding to

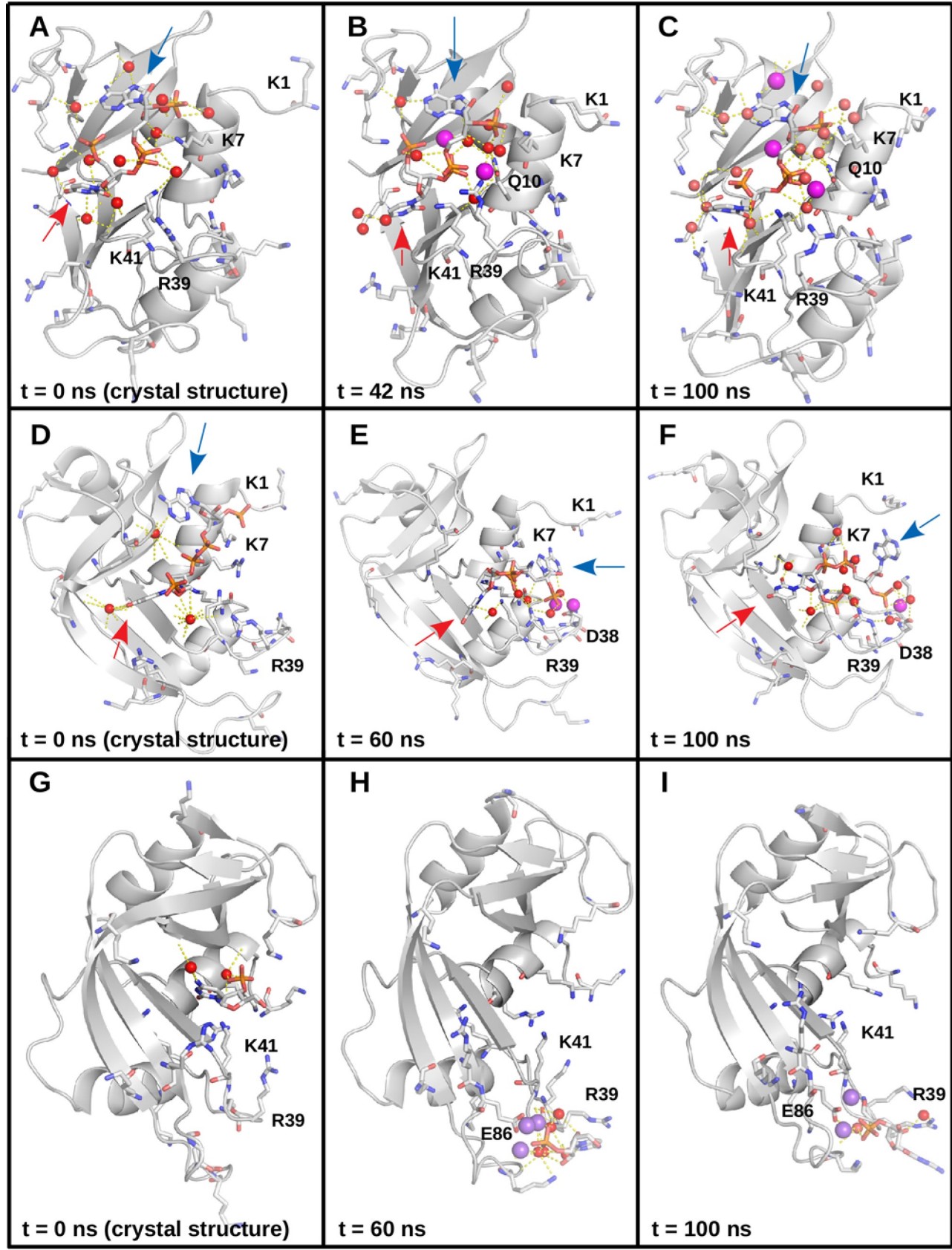

**Fig 7. Comparison between crystal and simulated structures.** (A) The RNase A–PUA complex from a crystal structure. The protein backbone is in cartoon representation, colored in gray. Lysines, arginines, and the ligand are given in stick representation with carbon atoms in white, nitrogens in blue, oxygens in red, and phosphorus in orange. Certain lysine and arginine residues are labeled. Water oxygens are represented as red spheres, polar contacts are shown as dashed lines. The locations of the urydyl and adenosyl moieties are indicated with a red and a blue arrow, respectively. (B) The RNase A–PUA complex after 42 ns of production dynamics. The protein backbone is in cartoon representation, colored in gray. Lysines, arginines, Q10, and the ligand are given in stick representation with carbon atoms in white, nitrogens in blue, oxygens in red, and phosphorus in orange. Certain lysine and arginine residues and Q10 are labeled. Water oxygens are represented as red spheres; sodium ions are represented as magenta spheres; polar contacts are shown as dashed lines. The locations of the urydyl and adenosyl moieties are indicated with a red and a blue arrow, respectively. (C) The RNase A–PUA complex after 100 ns of production dynamics. The protein backbone is in cartoon representation, colored in gray. Lysines, arginines, Q10, and the ligand are given in stick representation with carbon atoms in white, nitrogens in blue, oxygens in red, and phosphorus in orange. Certain lysine and arginine residues and Q10 are labeled. Water oxygens are represented as red spheres; sodium ions are represented as magenta spheres; polar contacts are shown as dashed lines. The locations of the urydyl and adenosyl moieties are indicated with a red and a blue arrow, respectively. (D) The RNase A–PAX complex from a crystal structure. The protein backbone is in cartoon representation, colored in gray. Lysines, arginines, and the ligand are given in stick representation with carbon atoms in white, nitrogens in blue, oxygens in red, and phosphorus in orange. Certain lysine and arginine residues are labeled. Water oxygens are represented as red spheres, polar contacts are shown as dashed lines. The locations of the thymine and adenosyl moieties are indicated with a red and a blue arrow, respectively. (E) The RNase A–PAX complex after 60 ns of production dynamics. The protein backbone is in cartoon representation, colored in gray. Lysines, arginines, and the ligand are given in stick representation with carbon atoms in white, nitrogens in blue, oxygens in red, and phosphorus in orange. Certain lysine and arginine residues are labeled. Water oxygens are represented as red spheres; sodium ions are represented as magenta spheres; polar contacts are shown as dashed lines. The locations of the thymine and adenosyl moieties are indicated with a red and a blue arrow, respectively. (F) The RNase A–PAX complex after 100 ns of production dynamics. The protein backbone is in cartoon representation, colored in gray. Lysines, arginines, and the ligand are given in stick representation with carbon atoms in white, nitrogens in blue, oxygens in red, and phosphorus in orange. Certain lysine and arginine residues are labeled. Water oxygens are represented as red spheres; sodium ions are represented as magenta spheres; polar contacts are shown as dashed lines. The locations of the thymine and adenosyl moieties are indicated with a red and a blue arrow, respectively. (G) The RNase A–C3P complex from a crystal structure. The protein backbone is in cartoon representation, colored in gray. Lysines, arginines, and the ligand are given in stick representation with carbon atoms in white, nitrogens in blue, oxygens in red, and phosphorus in orange. Certain lysine and arginine residues are labeled. Water oxygens are represented as red spheres, polar contacts are shown as dashed lines. (H) The RNase A–C3P complex after 60 ns of production dynamics. The protein backbone is in cartoon representation, colored in gray. Lysines, arginines, and the ligand are given in stick representation with carbon atoms in white, nitrogens in blue, oxygens in red, and phosphorus in orange. Certain lysine and arginine residues are labeled. Water oxygens are represented as red spheres; sodium ions are represented as magenta spheres; polar contacts are shown as dashed lines. (I) The RNase A–C3P complex after 100 ns of production dynamics. The protein backbone is in cartoon representation, colored in gray. Lysines, arginines, and the ligand are given in stick representation with carbon atoms in white, nitrogens in blue, oxygens in red, and phosphorus in orange. Certain lysine and arginine residues are labeled. Water oxygens are represented as red spheres; sodium ions are represented as magenta spheres; polar contacts are shown as dashed lines.

RNase A. Moreover, the complete lack of sodium ions, bridging or otherwise, in the 22 published structures we have used to perform molecular dynamics, taken together with the presence of citrate or sulfate ions in some of the structures, and the difficulty of distinguishing $Na^+$ from water in crystallography [51,52], suggests that the number of sodium ions in the structural databases may be underestimated.

Our work demonstrates that apart from the long-lived, high residence time bridging interactions, there exist 2–3 orders of magnitude more abundant short-lived ($< 1$ ns) bridging interactions. We show that residence times follow a power law distribution–there exist a single-digit number of bridging interactions with residence times on the order of tens of nanoseconds, tens of bridging interactions with lifetimes between 1 and 10 ns, and hundreds to thousands of such interactions with residence times below 1 ns. In this respect, bridging solvent behaves similarly to hydrating solvent [53,54].

Crucially, a simple, semiquantitative consideration of bridging solvent shows that for many RNase ligands, the energetic contribution to binding of the short-lived bridges is comparable or greater than that of the long-lived bridges. A convenient example is provided by the RNase A–C3P system where in replica 2, all bridging interactions have lifetimes below 1 ns, whereas replica 4 has multiple long-lived bridging interactions– 3 above 10 ns and 18 between 1 and 10 ns. However, the MM-GBSA and MM-PBSA results for replica 2 are lower than for replica 4 (-54.7 and -40.7 kcal/mol versus -19.7 and -31.6 kcal/mol, respectively.) While this is a crude, semiquantitative treatment of the matter (as other differences in between replicas may also influence the computed results), it is useful in highlighting that long-lived bridges may not be the dominant energetic contributor. Indeed, long residence times do not indicate particularly

strong protein–water or water–ligand interactions, but rather a topography that prevents the water molecule from exchanging by a cooperative mechanism [55], such as exchange-mediated orientational randomization (EMOR) at high confinement [56]. Therefore, preserving bridging interactions from crystallographic structures in a drug design campaign targeting a particular protein or protein–protein interaction [57] might not be sufficient. One also needs to account for short-lived solvent bridges. It is likely that molecular dynamics can be a powerful tool in this regard. This line of reasoning is further corroborated by the observation that a strong, positive correlation ($R^2 = 0.55$, see S1 File) exists between ligand potency and the average number of bridging molecules, observed in our simulations.

The $\Delta H$/p$K$ plots excluding solvent show no significant correlation. Moreover, the lines of best fit exhibit greatly differing slopes, i.e. they cross each other at large angles. Conversely, the corresponding plots with bridging solvent tend to exhibit significant correlations and smaller angles in between the lines of best fit. These findings can be interpreted as suggesting that the calculations without solvent omit the key factors, determining RNase–ligand binding, producing random, largely orthogonal patterns. Conversely, the calculations with bridging solvent include in themselves most of the determinants governing binding, and consistently produce reliable, reproducible patterns.

The variations in $\Delta H$ and $R^2$ across replicas indicate that the individual 100 ns simulations do not visit all conformations, relevant to binding. This is likely to be particularly pronounced for the pyrophosphate (or, equivalently, diphosphate) series of ligands, which also contact the protein's unstructured regions, including the highly flexible N-terminal loop (see Figs 1, 2, and S2). Indeed, out of all lysine and arginine residues, K1 exhibits the greatest rotamer diversity from the 22 published structures (Fig 2). For loops, a great number of conformational states is possible, especially for terminal loops, which are bound only on one end and have vastly more degrees of freedom than loops which are bound on both ends. During the simulations, we observed conformations where K1 contacts the ligands and others where it is far removed from them. Moreover, we observed conformations where K7, R39, and K41 contact the ligands, as well as conformations where they do not (Fig 7). Finally, although more distant residues, such as K66 and R85 (see Fig 2) did not participate substantially in ligand binding in our simulations, we cannot rule out that they are involved *in vivo*, where sampling of conformational space is not limited to 400 ns. Indeed, this is a vast conformational space we cannot hope to sample exhaustively. It has previously been demonstrated that in such a scenario, performing several shorter replicas is far more efficient in terms of sampling the relevant conformational space than performing a lesser number of longer replicas [58]. This particular system offers another, highly illustrative example of why this is the case. Some of the smaller ligands, being more mobile, tended to leave the binding cleft near the end of the simulations in certain replicas. In order to observe multiple binding and unbinding events for these complexes, we would need to increase the length of our simulations by at least an order of magnitude, likely two or more, which is presently not feasible. Therefore, a more efficient strategy to obtain different conformations, relevant to protein–ligand binding, is to perform several shorter replicas, which is the approach we have adopted. Indeed, although the $\Delta H$ and $R^2$ values vary in between replicas, $R^2$ consistently points to a significant correlation between affinity and bridging interactions, indicating that this is the case throughout all of conformational space. In other words, the strong correlations and the reproducible patterns we observe in our simulations across a wide spectrum of affinity and in four independent replicas indicate that our results and conclusions are robust.

It is noteworthy that the greatest outliers in the plots in Fig 6C and 6D are (pyro)phosphate-containing ligands which during dynamics make extensive amine–phosphate contacts with the protein through its arginine and lysine residues. Indeed, RNase A is abundant in such

residues, mostly within contact distance of the ligands around the binding groove (Fig 2). For example, in Fig 6C, C5P and A3P consistently lie well below the lines of best fit, meaning that their affinity for RNase A is greatly overestimated, eroding overall $R^2$. As previously demonstrated, these ligands make little direct amine–phosphate contacts with the protein in the crystal structures, but reorient themselves to make extensive contacts of this nature during molecular dynamics (Fig 4). Moreover, examining the per-residue energies reveals that lysine and arginine residues are indeed the greatest contributors to binding (Fig 8).

The only negative contribution to binding affinity is made by acidic residues in the vicinity of the phosphates, whereas the greatest positive contributions come from arginines and lysines, clustered together with the phosphates (compare Figs 4 and 8), highlighting once more the key role of the phosphates. We also note the importance of salt-linked triads to protein–ligand

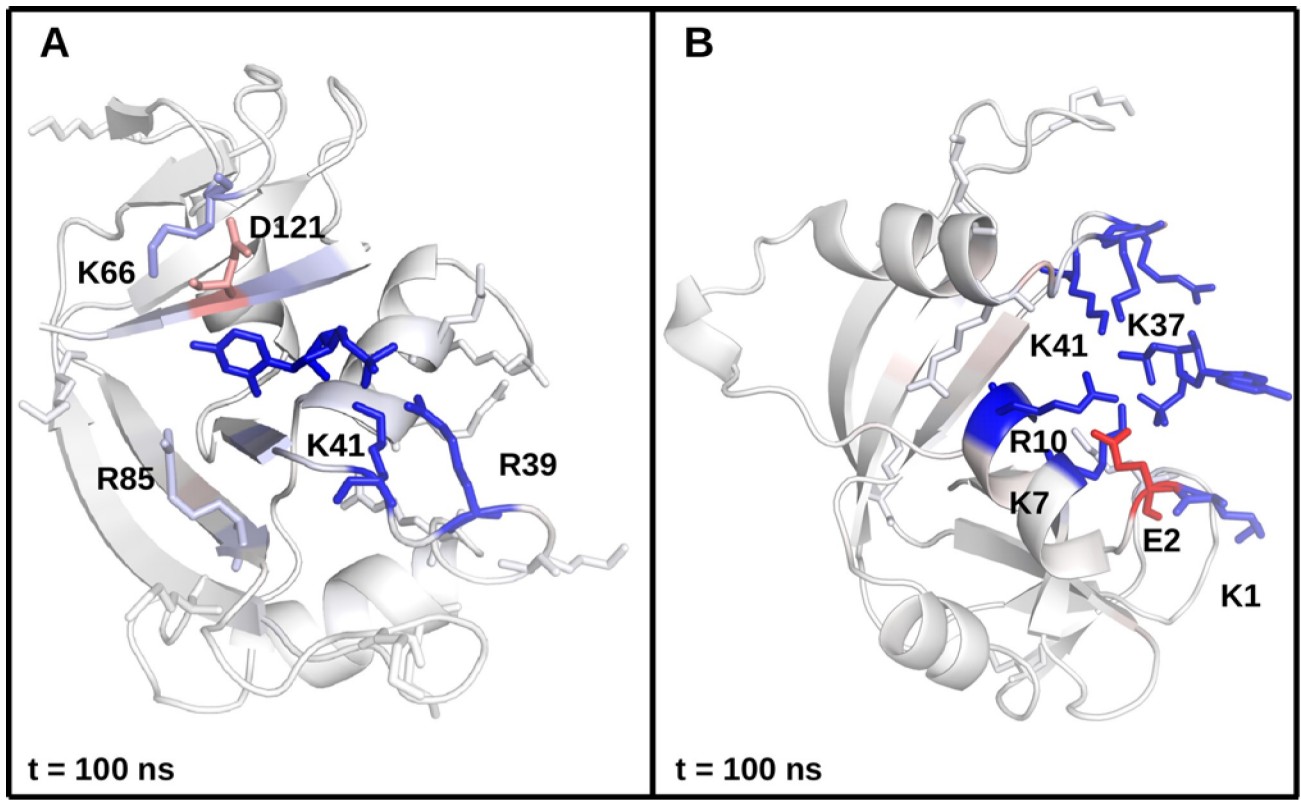

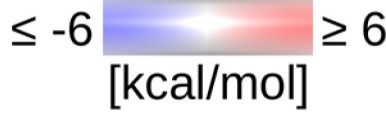

**Fig 8. Sources of affinity assessed via energetics analysis based on protein–ligand complex trajectories.** (A) The RNase A–C5P complex after 100 ns of molecular dynamics colored by per-residue energies of interaction calculated from the entire trajectory. Blue indicates a favorable contribution to binding, red indicates an unfavorable contribution. R39, K41, K66, R85, and D121 are shown in stick representation and explicitly labeled; the ligand is also shown in stick representation. Note the frame in the image is the same as in Fig 4B. (B) The RNase A–A3P complex after 100 ns of molecular dynamics colored by per-residue energies of interaction calculated from the entire trajectory. Blue indicates a favorable contribution to binding, red indicates an unfavorable contribution. K1, E2, K7, R10, K37, and K41 are shown in stick representation and explicitly labeled; the ligand is also shown in stick representation. Note the frame in the image is the same as in Fig 4D.

binding in this system, as this is a cooperative interaction that has previously been shown to be involved in protein folding [42,43] and protein–protein recognition and binding [44].

Our structural and energetics analysis suggest that the overestimated ΔH values stem from the overly attractive amine–phosphate potential–a previously recognized problem in the CHARMM and Amber series of force fields [59]. Indeed, given the abundance of amine-containing side chains on the protein side and the abundance of phosphates on the ligand side, RNase–ligand simulations are likely to be heavily burdened by the inaccuracies in the parameterization of the amine–phosphate interaction. We, therefore, applied NBFIX (nonbonded fix) corrections by increasing the σ parameter in the Lennard-Jones potential for the N–O = P (amine nitrogen–phosphate oxygen) pair [59,60]. NBFIX parameters are pair-specific corrections, i.e. they affect only a specific atom pair, e.g. amine nitrogen–carboxylate oxygen or amine nitrogen–phosphate oxygen. The NBFIX corrections represent an increase in the σ parameter for a given atom pair, making it less attractive. We also applied NBFIX corrections to the CT–CT atom pair (the interaction between two alyphatic carbons) [61], as these have also been shown to be overly stabilizing, artificially reducing the radius of gyration of simulated proteins in comparison to experiment [62], as well as NBFIX corrections to the $Na^+$–$Cl^-$ (sodium ion–chloride ion) and $Na^+$–O = P (sodium ion–phosphate oxygen) pair [63]. After performing analogous analysis on the systems simulated with NBFIX corrections, we found that compounds that lack phosphate groups (TXS, U1S, U2S, 0G0, 0FT, and 0EY) exhibit very similar ΔH values to their respective non-NBFIX behavior (compare S6A and S6C and S6B and S6D Fig, note that the Y axis scales are different; also see S1 File), confirming that the amine–phosphate interactions are the dominant difference in between the NBFIX and standard parameter simulations. For the phosphate-containing compounds, the modified potential resulted in a reduced propensity for direct amine–phopshate contacts during production dynamics. For the pyrophosphate derivatives, this also resulted in a decrease in the number of salt-linked triads and clusters formed by phosphate oxygens and positively charged side chains. This corresponded to an upshift in ΔH values for the phosphate-containing compounds, as compared to the non-NBFIX simulations. The NBFIX corrections, however, reduced $R^2$, rather than increase it. This implies that the previously published parameters [59,60] are unsuitable for the current protein–ligand data set, highlighting the problem of parameter transferability in molecular dynamics. Parameterizing the vast chemical space that drug and drug-like molecules constitute is certainly a daunting task. A recent analysis on the charging and dispersive-repulsive contributions to solvation free energies in an analysis of GAFF and OPLSA-AA with the TIP3P, SPCE, and OPC3 water models has demonstrated that while GAFF generally performs well, further improvements are more likely to be obtained through "adjusting and tuning the available atomic charge calculation protocols, namely AM1/BCC for GAFF2 and 1.14*CM1A or 1.14*CM1A-LBCC for OPLS-AA" [64]. Our extensive, fully independent study complements this work by examining a different property in a different system, leading us to concur with Vasetti et al. [64].

Finally, we discuss two methodologically important points. First, we have opted to use a relatively large solvation shell (15 Å instead of the usual 10–11–12) because we noticed an imaging artifact in certain trajectories when using smaller solvation shells where bridging interactions could not be detected by *cpptraj*, leading to an underestimation of the magnitude of ΔH. Second, we point out that the computationally much cheaper MM-GBSA calculations offer similar performance to the more theoretically rigorous MM-PBSA approach in terms of $R^2$ using the standard Amber parameters (Fig 6), despite being a much cruder approximation, at a lower level of theory. Thus, in scenarios with limited computational resources, particularly limited CPU resources, MM-GBSA offers a viable alternative to the more demanding (typically, 1–2 orders of magnitude in terms of compute time) MM-PBSA analysis.

## Conclusions

We have performed and presented extensive free energy calculations on a protein–ligand data set spanning 22 compounds and nearly five orders of magnitude in affinity. This work presents an independent test of the state-of-the-art fixed-charge force fields. Crucially, we describe an automated workflow for the detection of bridging interactions in protein–ligand binding–an often important but neglected factor in intermolecular interactions. Moreover, our workflow is extensible and amenable to modification to accommodate other types of biologically important (macro)molecules such as nucleic acids, saccharides, lipids, and polyamines, as well as multicomponent systems of these, where complex, multifactor bridging interactions are likely to be crucial for an accurate, atomic-level description of the biology of interest [65,66]. The workflow we describe is also likely to be applicable to four-dimensional, time-dependent quantitative-structure activity relationship (4D-QSAR) studies [45] in drug design and optimization.

## Supporting information

**S1 Fig. Cα RMSDs.** (A) Cα RMSDs for the protein over the course of production dynamics in replica 1. (B) Cα RMSDs for the protein over the course of production dynamics in replica 2. (C) Cα RMSDs for the protein over the course of production dynamics in replica 3. (D) Cα RMSDs for the protein over the course of production dynamics in replica 4.
(EPS)

**S2 Fig. Cα RMSFs.** (A) Cα RMSFs for the protein over the course of production dynamics in replica 1. (B) Cα RMSFs for the protein over the course of production dynamics in replica 2. (C) CαRMSFs for the protein over the course of production dynamics in replica 3. (D) Cα RMSFs for the protein over the course of production dynamics in replica 4.
(EPS)

**S3 Fig. Ligand heavy atom RMSDs.** (A) Ligand heavy atom RMSDs over the course of production dynamics in replica 1. (B) Ligand heavy atom RMSDs over the course of production dynamics in replica 2. (C) Ligand heavy atom RMSDs over the course of production dynamics in replica 3. (D)Ligand heavy atom RMSDs over the course of production dynamics in replica 4.
(EPS)

**S4 Fig. Enthalpy convergence without bridging solvent.** (A) Enthalpy (ΔH) convergence from the MM-PBSA calculations from replica 1 (top), replica 2, replica 3, and replica 4 (bottom); bridging waters and ions are not included in the calculations. (B) Enthalpy (ΔH) convergence from the MM-GBSA calculations from replica 1 (top), replica 2, replica 3, and replica 4 (bottom); bridging waters and ions are not included in the calculations.
(PDF)

**S5 Fig. Enthalpy convergence with bridging solvent.** (A) Enthalpy (ΔH) convergence from the MM-PBSA calculations from replica 1 (top), replica 2, replica 3, and replica 4 (bottom); bridging waters and ions are included in the calculations. (B) Enthalpy (ΔH) convergence from the MM-GBSA calculations from replica 1 (top), replica 2, replica 3, and replica 4 (bottom); bridging waters and ions are included in the calculations.
(EPS)

**S6 Fig. ΔH/p*K* correlations with NBFIX parameters.** (A) Correlations between computed enthalpy (ΔH) and experimental affinity from the MM-PBSA calculations from replica 1 (blue), replica 2 (red), replica 3 (green), and replica 4 (black) with NBFIX corrections; bridging

waters and ions are included in the calculations. For each replica, the line of best fit and the corresponding equation are also given in the respective color. (B) Correlations between computed enthalpy ($\Delta H$) and experimental affinity from the MM-GBSA calculations from replica 1 (blue), replica 2 (red), replica 3 (green), and replica 4 (black) with NBFIX corrections; bridging waters and ions are included in the calculations. For each replica, the line of best fit and the corresponding equation are also given in the respective color.
(EPS)

**S1 File. Supplementary data.** Sheet 1 contains the experimental affinity data between RNase A and the 22 ligands, expressed as $pK_i$ values, and the computed enthalpies from the four replicas. Sheet 2 contains the average C$\alpha$ RMSD values; sheet 3 –the C$\alpha$ RMSFs; sheet 4 –the ligand heavy atom RMSDs; sheet 5 –the convergence of the MM-PBSA and MM-GBSA values without bridging solvent; sheet 6 –the convergence of the MM-PBSA and MM-GBSA values with bridging solvent; sheet 7 –the average number of bridging waters and ions across the four replicas for every compound and the corresponding standard deviation, followed by the average number of bridges and the standard deviations; sheet 8 –the radial distribution functions for bridging water and bridging sodium ions around the ligands calculated from the four replicas.
(XLSX)

## Author Contributions

**Conceptualization:** Stefan M. Ivanov.

**Data curation:** Stefan M. Ivanov.

**Formal analysis:** Stefan M. Ivanov, Ivan Dimitrov, Irini A. Doytchinova.

**Funding acquisition:** Stefan M. Ivanov, Ivan Dimitrov, Irini A. Doytchinova.

**Investigation:** Stefan M. Ivanov.

**Methodology:** Stefan M. Ivanov.

**Project administration:** Ivan Dimitrov, Irini A. Doytchinova.

**Resources:** Stefan M. Ivanov.

**Software:** Stefan M. Ivanov.

**Supervision:** Stefan M. Ivanov.

**Validation:** Stefan M. Ivanov.

**Visualization:** Stefan M. Ivanov.

**Writing – original draft:** Stefan M. Ivanov.

**Writing – review & editing:** Stefan M. Ivanov, Ivan Dimitrov, Irini A. Doytchinova.

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
