## [Decision Letter · Decision Letter 0]

11 Sep 2019

[EXSCINDED]

PONE-D-19-23148

Bridging solvent molecules mediate RNase A – ligand binding

PLOS ONE

Dear Dr Ivanov,

Thank you for submitting your manuscript to PLOS ONE. After careful consideration, we feel that it has merit but does not fully meet PLOS ONE’s publication criteria as it currently stands. Therefore, we invite you to submit a revised version of the manuscript that addresses the points raised during the review process.

The two reviewers combine for 16 important points. While some of them are minor, most of them are significant enough to make me unable to determine the validity of this work. 

I encourage you to address all of the points so that we can fully assess this manuscript.

We would appreciate receiving your revised manuscript by Oct 26 2019 11:59PM. To enhance the reproducibility of your results, we recommend that if applicable you deposit your laboratory protocols in protocols.io, where a protocol can be assigned its own identifier (DOI) such that it can be cited independently in the future. For instructions see: http://journals.plos.org/plosone/s/submission-guidelines#loc-laboratory-protocols

We look forward to receiving your revised manuscript.

Kind regards,

Freddie Salsbury , Jr, PhD

Academic Editor

PLOS ONE

Journal Requirements:

Reviewers' comments:

Reviewer's Responses to Questions

**Comments to the Author**

1. Is the manuscript technically sound, and do the data support the conclusions?

Reviewer #1: Partly

Reviewer #2: Partly

2. Has the statistical analysis been performed appropriately and rigorously? 

Reviewer #1: No

Reviewer #2: Yes

3. Have the authors made all data underlying the findings in their manuscript fully available?

Reviewer #1: No

Reviewer #2: Yes

4. Is the manuscript presented in an intelligible fashion and written in standard English?

Reviewer #1: Yes

Reviewer #2: Yes

5. Review Comments to the Author

Reviewer #1: The authors presented a MD study for RNase A with different ligand

compounds using the AMBER force field. Their key result includes

bridging water analysis, heavy atoms RMSD, and energetic analysis for

the ligands with and without water and ions. While the outcome that

inclusion of bridging waters lead to higher correlations with

experimental data is useful there are signficant problems with the

manuscript. Comments follow.

1) Importanly, the authors need to discuss published experimental

structural studies in relation to their work validation beyond two

lines on page 15. Based on text on page 2 (references 8 to 11) there

is experimental information available.

2) In Figure 1 the phosphates of the ligands are shown to be fully

protonated (ie. neutral). If this is the protonation state used in

the simualtions, then the results are rather meaningless as this state

will not exist in the vicinity of pH 7.0.

3) What is the need to use the modified parameters for additional 8.8 or

10 microseconds? What are the modified parameters and to they impact

the results at all. This is not discussed in the methods, but appears

to be associated with the use of NBFIX parameters that is presented

later in the manuscript. This issue needs significant clarification.

4) The MM-PBSA/GBSA as presented as enthalpies of binding; however, PB

and GB are giving the free energies of solvation, such that using free

energy of binding may be more appropraite. This needs to be

clarified.

5) The authors need to perform some structural analysis (maybe

spatial/angular/radial distribution) where the bridging functional

groups between protein and the ligand are involved. Or role of the

phosphate group in the binding mechanism of salt bridges

(phosphate-lysine etc). The detailed structural analysis will show the

importance of those functional groups that get involved in the most

likely bridging through the entire trajectory.

6) On page 4 the authors state "Average Cα RMSD is 0.07 Å." This is

clearly incorrect based on the RMSD figures

7) The RMSD/RMSF figures should be moved to the Supplementary Material

8) Figures 7 and 8 may also be moved to the Supplementary Material.

It would be nice to provide some interpretation in relation to Figure

6. Why a certain fragment has less or more number of bridges in

comparison to another ligand? Also, some more details/discussion in

the calculation to the lifetimes of the ligand or all the all bound

for the duration of the simulations. Please clarify.

9) Clarify the scientific information derived from Figures 9? We know the

importance of water and ions in the formation of the bridges. How

about having a somewhat larger variation in the DH for a different

replica in particularly for PAX, PUA, PAP (Figure 9C and D)?

10) Indeed, the Figures 9 and 10 should be moved to the Supplementary

Material and the results included in the main text as a table. This

will make it much easier for the reader to access those results. This

is important as the main result is the improved correlation with

experiment when water is included in the enthalpy of binding

calculation.

11) The following needs some structural analysis to support the statement:

“the plots in Fig 9C and D, and the molecular dynamics trajectories

reveals that the greatest outliers are phosphate or pyrophosphate

containing molecules whose interactions with the protein are dominated

by phosphate – lysine or phosphate – arginine interactions.”

Reviewer #2: The manuscript PONE-D-19-23148 by Stefan Ivanov et al. studies the water bridging interactions in RNase-ligand binding using explicit molecular dynamics simulations. Each of the 22 compounds bound to RNase A is simulated in four independent replicas with a length of 100 ns for each replica. Some analyses are performed on bridging water molecules to show the importance of their role in RNase-ligand binding. MM-PBSA, MM-GBSA are used to compute the binding energies of the compounds. In order to explain the overestimated dH values, the same systems are simulated with nonbonded fix (NBFIX) corrections to the force fields. And similar analyses are performed on the second round of simulations. However, there are a number of issues that I hope the authors could address.

Major points are as follows:

1. One advantage of explicit solvent MD simulation is to provide structural information at atomic resolution. However, I couldn't see any detailed structural information of the bridging water molecules in the binding site. For example, where are the water molecules in the binding sites? Is there any hydration site in the binding pocket that is critical for the binding? What are the configurations of the residues, ligands and water molecules in the binding site? Overall, I think the authors should include more detailed discussions about bridging interactions with atomic structures.

2. In the paragraph starting from line 186, "As we were primarily interested in relative binding energies ...", entropy calculations are omitted, however, I don’t think the assumption that the entropy changes are similar across different systems in this work is valid without any proof. Some of the 22 compounds simulated in this work (Fig 1) are quite different from each other in terms of rotational bonds, molecular weights. The water molecule involved in the bridging interactions may also have a considerable contribution to the entropy. Could you show any validation that the entropy is negligible in computing the relative free energies?

3. Although the enthalpy changes computed in the manuscript are related to the relative binding energies, they are not the same. Could you explain more about the choice of computing dHs instead of computing the ddGs?

4. I am puzzled by the plots in Fig. 9 and Fig. 10, which is trying to correlate between the computed relative binding energies and the absolute experimental binding affinities. Shouldn’t the correlation be between the computed ddG vs experimental ddG?

5. I also have concerns about the statistical stability of computed binding free energies. There are large variations between the dHs computed from four different replicas and the slops of fitted linear functions in Fig. 9 and Fig. 10 also show large variations, which may indicate that the conformational samplings from the simulations aren’t enough for the binding energy calculations.

Some minor points:

1. Line 179, “MM-PBA” should be “MM-PBSA”

2. In Fig. 1, the pKi value of U2S is in the wrong place.

3. Reference 56 is in the wrong line.

6. PLOS authors have the option to publish the peer review history of their article (what does this mean?). If published, this will include your full peer review and any attached files.

Reviewer #1: No

Reviewer #2: No

---

## [Author Response · Author response to Decision Letter 0]

24 Sep 2019

We thank the reviewers for their positive and constructive comments. Below, we provide a point-by-point response to each of their concerns.

Reviewer #1: The authors presented a MD study for RNase A with different ligand compounds using the AMBER force field. Their key result includes bridging water analysis, heavy atoms RMSD, and energetic analysis for the ligands with and without water and ions. While the outcome that inclusion of bridging waters lead to higher correlations with experimental data is useful there are signficant problems with the manuscript. Comments follow.

1) Importanly, the authors need to discuss published experimental structural studies in relation to their work validation beyond two lines on page 15. Based on text on page 2 (references 8 to 11) there is experimental information available.

We thank the reviewer for his comment, which we feel is very constructive. Moreover, it aligns nicely with several other comments by both reviewers. Accordingly, as requested by the reviewer, we have included structural analysis on the behavior of several complexes throughout production dynamics. We believe the added details, along with further structural and analysis details we have included throughout the manuscript, will be highly instructive for the PLOS ONE readership. More details are provided below.

2) In Figure 1 the phosphates of the ligands are shown to be fully protonated (ie. neutral). If this is the protonation state used in the simualtions, then the results are rather meaningless as this state will not exist in the vicinity of pH 7.0.

The reviewer is indeed correct to point out that the phosphate groups will not be protonated at pH 7. Accordingly, in our original submission, the last line of the Figure 1 caption explicitly states that the phosphate groups have been drawn protonated, but have been simulated in a fully deprotonated state - “Note that phosphate groups are drawn protonated, whereas they have been simulated in a fully deprotonated state. ” We have deliberately placed this sentence at the very end of the caption to make it stand out. In our revised manuscript, the Figure 1 caption remains unchanged. We thank the reviewer for pointing out that this information can be hard to notice. Therefore, to help guide the reader, we have added this notice in the main text as well. 

3) What is the need to use the modified parameters for additional 8.8 or 10 microseconds? What are the modified parameters and to they impact the results at all. This is not discussed in the methods, but appears to be associated with the use of NBFIX parameters that is presented later in the manuscript. This issue needs significant clarification.

The NBFIX parameters are pair-specific corrections to the potential energy function. It has previously been noted by other authors that certain interactions in the CHARMM and AMBER force fields are significantly overestimated. For example, if glycine is simulated in water, it will aggregate in simulations, whereas in reality it is highly soluble. This is because the amine – acetate interaction between the N-terminal amine and the C-terminal COO- group is too attractive in the current force fields. This is corrected by increasing the σ (sigma) parameter in the potential energy function for the amine nitrogen (N) – carboxylate oxygen (O) atom pair. More details about NBFIX parameters and parametrization can be found in the papers we reference. We have included simulations with NBFIX corrections because testing how general and useful they are is of great interest to the drug design and computational chemistry community. We have 22 complexes, for each of which we have performed 400 ns (4 replicas x 100 ns) of production dynamics = 8.8 microseconds in total. For a fair comparison, we have performed the exact same amount of sampling with NBFIX corrections, otherwise any comparison might be misleading. Whether or not the NBFIX parameters are applicable to our data set, however, is of secondary importance in this work; the main focus is on bridging interactions. Therefore, we briefly discuss NBFIX in the Discussion section. We have chosen not to include the NBFIX simulations in the Methods section, but to simply state that we have performed the same type of analysis on simulations performed with NBFIX corrections, in order to maintain the focus of this work on bridging interactions. We thank the reviewer for pointing out this can be seen as a little unclear. We have added more details and explanations on the NBFIX work to make this section clearer and easier to digest.

4) The MM-PBSA/GBSA as presented as enthalpies of binding; however, PB and GB are giving the free energies of solvation, such that using free energy of binding may be more appropraite. This needs to be clarified.

Polar solvation free energies are obtained by solving the linearized Poisson-Boltzmann equation or the Generalized Born equation. This is where the “PB” and “GB” in MM-PBSA and MM-GBSA come from. However, Poisson-Boltzmann and Generalized Born calculations are not the only calculations in MM-PBSA and MM-GBSA – they also include surface area (or molecular volume) calculations for nonpolar solvation, gas phase energies, and van der Waals terms, calculated in accord with the force field parameters. Combined, these calculations are known as molecular mechanics – Poisson-Boltzmann surface area (MM-PBSA) and molecular mechanics – Generalized Born surface area (MM-GBSA) calculations. MM-PBSA and MM-GBSA, by themselves, produce the enthalpy of interaction (DH). In order to obtain the free energy of interaction (DG), as opposed to the enthalpy, one also needs to calculate the entropy of interaction (DS). This, however, is not possible at present, perhaps except for the very simplest of model systems, which is certainly not the case with our data set. Although calculations such as normal mode analysis (NMA) are sometimes performed, they are too expensive and too unreliable to obtain accurate DGs. Therefore, we focus our attention on the only part of the binding free energy that can be calculated with a certain amount of reliability – enthalpy (DH). Entropy calculations await further methodological and computational developments. More details on MM-PB(GB)SA can be found in the papers we reference. 

5) The authors need to perform some structural analysis (maybe spatial/angular/radial distribution) where the bridging functional groups between protein and the ligand are involved. Or role of the phosphate group in the binding mechanism of salt bridges (phosphate-lysine etc). The detailed structural analysis will show the importance of those functional groups that get involved in the most likely bridging through the entire trajectory.

We thank the reviewer for this comment, as we feel it is particularly constructive and has led to interesting insights. We have performed additional analysis on the bridging waters and ions, examining their radial distribution around the ligands. We find that short-lived bridging is dominated by water, whereas long-lived bridging is dominated by sodium ions, which interact mostly with the ligands’ phosphate groups and oxygen atoms from protein side chains, as well as backbone oxygens. Moreover, the complete lack of bridging chloride ions (Cl-) in our entire data set clearly points to the preeminence of the phosphates in RNase – ligand binding. Moreover, even compounds that lack phosphate groups do not employ bridging Cl-, only Na+ and water, although, typically, to a lesser extent than phosphate-containing compounds. We have included more structural detail and discussion on this to the Results and Discussion sections.

6) On page 4 the authors state "Average Cα RMSD is 0.07 Å." This is clearly incorrect based on the RMSD figures

We thank the reviewer for pointing out that this figure and the caption information can be somewhat confusing. The figure presents the starting crystal and NMR structures we have used to perform MD. The 0.07 Å number for the Cα RMSD is given for the RMSD between the starting crystal and NMR structures, not the average RMSD during dynamics. The 0.07 number can easily be verified by loading chain A from PDB structures 4g90, 1rnm, 4g8y, 3d6p, 1w4p, 1jvu, 1qhc, 1jn4, 1w4o, 1afl, 1o0f, 1afk, 3d6o, 1u1b, 1o0h, 1o0n, 4g8v, 1o0m, 3d8z, 1z6s, 1rpf, 1w4q in Pymol and aligning everything to 4g90 – the output is 0.073 Å for average Cα RMSD. In order to avoid confusion, we have removed the number from the caption and have modified the figure title.

7) The RMSD/RMSF figures should be moved to the Supplementary Material

As suggested by the reviewer, these and several other figures have been moved to SI. We thank the reviewer for this suggestion, as this has opened up more space in the manuscript for more figures and discussion on structural analysis, which has improved the quality of the manuscript.

8) Figures 7 and 8 may also be moved to the Supplementary Material. It would be nice to provide some interpretation in relation to Figure 6. Why a certain fragment has less or more number of bridges in comparison to another ligand? Also, some more details/discussion in the calculation to the lifetimes of the ligand or all the all bound for the duration of the simulations. Please clarify.

As suggested by the reviewer, we have moved these figures to SI. In order to clarify what we mean by “bridging molecule” and “bridging interaction,” we have added more illustrative examples in the relevant Methods section. In the Results section, we note that phosphate-containing ligands tend to attract more bridging waters and ions than compounds that do not contain phosphates, and that the large, pyrophosphate-contining compounds attract more bridging waters than the small compounds that contain only one phosphate group. We note that a pyrophosphate (or diphosphate) group is simply two phosphate groups condensed together. It becomes clear, then, that phosphates tend to attract more bridging waters and ions. The more such waters and ions become involved, the larger the complexes become, and the longer it takes for the relevant MM-PBSA and MM-GBSA calculations to converge. This is clarified in the Results section. 

9) Clarify the scientific information derived from Figures 9? We know the importance of water and ions in the formation of the bridges. How about having a somewhat larger variation in the DH for a different replica in particularly for PAX, PUA, PAP (Figure 9C and D)?

This comment aligns nicely with comment #5 from Reviewer 2. The variations in DH and R2 indicate that the individual simulations are not long enough to sample all conformations relevant to binding. Indeed, as we point out in the manuscript, some of the ligands contact the protein’s unstructured regions, which also bear lysine and arginine residues, especially the N-terminal loop, which bears lysine 1. Because this loop is free on the N terminus, it has a vast number of degrees of freedom, far more than the other loops, which are bound on both ends. This is reflected in the RMSF values of the loops. In fact, K1 has the greatest root-mean-square fluctuations of all residues, as evident from the relevant figure. In some conformations, K1 contacts the ligands, in others it is far removed from them. The same applies to many of the other lysines and arginines in the structure, e.g. K7, R39, K41, etc. We can not sample all possible conformations. In such a scenario, one should run multiple replicas to sample as much of conformation space as possible. Although there are variations in between replicas, the correlations are highly significant in every replica, which indicates that bridging waters are indeed a key determinant. Extended discussion of this has been included in the Discussion section.

10) Indeed, the Figures 9 and 10 should be moved to the Supplementary Material and the results included in the main text as a table. This will make it much easier for the reader to access those results. This is important as the main result is the improved correlation with experiment when water is included in the enthalpy of binding calculation.

As suggested by the reviewer, Figure 9 has been moved to SI. However, we feel that our data set is far too large to be included in the body of the manuscript in table form. We fully agree with the reviewer that the reader should have access to this data to examine it in detail. Therefore, we have included this data, along with more of our results, in a supplementary file the reader can download and examine in great detail.

11) The following needs some structural analysis to support the statement:

“the plots in Fig 9C and D, and the molecular dynamics trajectories reveals that the greatest outliers are phosphate or pyrophosphate containing molecules whose interactions with the protein are dominated by phosphate – lysine or phosphate – arginine interactions.”

As requested by the reviewer, we have performed ample structural and energetics analysis supporting our conclusions and have placed additional information on the relevant methods and discussion in the appropriate sections of the manuscript. Briefly, in our MM-PBSA and MM-GBSA calculations, similarly to our previously published work, we have enabled per-residue energy decompositions. This allows one to assess the contribution of each individual residue to binding during the molecular dynamics simulations. In other words, this analysis reveals whether a given residue makes a favorable or unfavorable contribution to binding or is largely indifferent. Together with our structural analysis, this confirmed that the overestimation in DH likely comes mostly from the inaccuracies in the parametrization of the amine – phosphate interaction. The relevant discussion has been placed in the Discussion section of the manuscript.

Reviewer #2: The manuscript PONE-D-19-23148 by Stefan Ivanov et al. studies the water bridging interactions in RNase-ligand binding using explicit molecular dynamics simulations. Each of the 22 compounds bound to RNase A is simulated in four independent replicas with a length of 100 ns for each replica. Some analyses are performed on bridging water molecules to show the importance of their role in RNase-ligand binding. MM-PBSA, MM-GBSA are used to compute the binding energies of the compounds. In order to explain the overestimated dH values, the same systems are simulated with nonbonded fix (NBFIX) corrections to the force fields. And similar analyses are performed on the second round of simulations. However, there are a number of issues that I hope the authors could address.

Major points are as follows:

1. One advantage of explicit solvent MD simulation is to provide structural information at atomic resolution. However, I couldn't see any detailed structural information of the bridging water molecules in the binding site. For example, where are the water molecules in the binding sites? Is there any hydration site in the binding pocket that is critical for the binding? What are the configurations of the residues, ligands and water molecules in the binding site? Overall, I think the authors should include more detailed discussions about bridging interactions with atomic structures.

This comment nicely aligns with several of the comments by Reviewer 1. We have included ample and illustrative structural information and analysis throughout the manuscript see above), which we believe would be highly instructive and very useful to the PLOS ONE readership.

2. In the paragraph starting from line 186, "As we were primarily interested in relative binding energies ...", entropy calculations are omitted, however, I don’t think the assumption that the entropy changes are similar across different systems in this work is valid without any proof. Some of the 22 compounds simulated in this work (Fig 1) are quite different from each other in terms of rotational bonds, molecular weights. The water molecule involved in the bridging interactions may also have a considerable contribution to the entropy. Could you show any validation that the entropy is negligible in computing the relative free energies?

First, we note that we are not performing relative free energy calculations (DDG) but enthalpy calculations (DH). The reviewer is indeed correct to point out that the ligands are significantly dissimilar and entropy cancelation should not be expected. Indeed, we chose to omit entropy calculations not because we believe that these terms will cancel out, but because at the given time it is impossible to calculate entropy reliably. It has been shown in previous work by us and others that enthalpies can be calculated reliably, but entropies can not. Given that entropy calculations are highly expensive but likely to introduce more noise than signal, we chose to not perform them. Moreover, given that we can demonstrate an important role for bridging waters – they key focus of this work – without entropy calculations, we feel that attempting to calculate entropy would be redundant. The phrase “As we were primarily interested in relative binding energies” pertains to the R2 values between the computed enthalpies, not the exact values of DH. Our 2016 paper shows that computed DH values are of similar magnitude to experimental DH results from ITC. This is explicitly referenced and discussed, as we feel that this is highly instructive for the reader. 

3. Although the enthalpy changes computed in the manuscript are related to the relative binding energies, they are not the same. Could you explain more about the choice of computing dHs instead of computing the ddGs?

This relates to the reviewer’s previous comment. Given that with MM-PB(GB)SA we can not reliably compute entropies, we are left with the enthalpies of interaction, i.e. it is impossible to reliably calculate DG with this technique. Although other techniques, such as thermodynamic integration (TI), circumvent the problem of explicit entropy calculation, they suffer from other methodological weaknesses, such as dependence of the computed DG on box size, ligand net charge (calculated DGs for charged ligands are highly inaccurate), etc. This is a significant limitation in computational chemistry that is widely discussed in the literature, partly in our 2019 paper (referenced in this manuscript) as well, and the references therein. Stated briefly – it is currently not possible to reliably calculate DG for large and complex ligands, especially charged ones – this is a well known shortcoming of MD and computational chemistry in general. Given that it is impossible to calculate DG reliably but that it is possible to calculate DH with reasonable accuracy, we focus our analysis on enthalpies (DHs) rather than binding free energies (DGs).

4. I am puzzled by the plots in Fig. 9 and Fig. 10, which is trying to correlate between the computed relative binding energies and the absolute experimental binding affinities. Shouldn’t the correlation be between the computed ddG vs experimental ddG?

Figures 9 and 10 correlate the enthalpy of binding (DH) between RNase A and the 22 ligands with experimental affinities. We calculate only DH because it is practically not possible to calculate DG for these compounds. We then relate computed DH values to experimental affinity data from the PDBbind database, which comes in the form of pKi values. While experimental DG can be obtained from the pKi values, there is no need to do so because we are not attempting to calculate DG. We stress that we are not attempting to correlate calculated with experimental DG because we have not calculated DG. 

5. I also have concerns about the statistical stability of computed binding free energies. There are large variations between the dHs computed from four different replicas and the slops of fitted linear functions in Fig. 9 and Fig. 10 also show large variations, which may indicate that the conformational samplings from the simulations aren’t enough for the binding energy calculations.

The reviewer is quite correct to point out that the variations in DH suggest that our simulations are not long enough to sample all relevant conformations – this is stated in our original submission as well. While performing longer simulations is certainly an option, it has been convincingly shown by others that more replicas offer much more efficient sampling than an equivalent amount of longer simulations. In our particular case, some of the smaller and more mobile ligands leave the binding cleft near the end of the simulations in some of the replicas. Doubling or even tripling the duration of the simulations will likely not be enough to observe multiple binding and unbinding events, which would require at least an order of magnitude more simulation time (likely several microseconds, at least). Therefore, a much more efficient strategy in this case is to perform several shorter simulations, rather that one long one, which is precisely the approach we have adopted. We have included more discussion on this in the manuscript, as well as a highly instructive reference the interested reader can examine. 

Some minor points:

1. Line 179, “MM-PBA” should be “MM-PBSA”

2. In Fig. 1, the pKi value of U2S is in the wrong place.

3. Reference 56 is in the wrong line.

We thank the reviewer for his careful examination of our work and for noticing these errors. These have now been corrected.

---

## [Decision Letter · Decision Letter 1]

10 Oct 2019

Bridging solvent molecules mediate RNase A – ligand binding

PONE-D-19-23148R1

Dear Dr. Ivanov,

We are pleased to inform you that your manuscript has been judged scientifically suitable for publication and will be formally accepted for publication once it complies with all outstanding technical requirements.

With kind regards,

Freddie Salsbury , Jr, PhD

Academic Editor

PLOS ONE

Additional Editor Comments (optional):

Reviewers' comments:

Reviewer's Responses to Questions

**Comments to the Author**

1. If the authors have adequately addressed your comments raised in a previous round of review and you feel that this manuscript is now acceptable for publication, you may indicate that here to bypass the “Comments to the Author” section, enter your conflict of interest statement in the “Confidential to Editor” section, and submit your "Accept" recommendation.

Reviewer #1: All comments have been addressed

Reviewer #2: All comments have been addressed

2. Is the manuscript technically sound, and do the data support the conclusions?

Reviewer #1: Yes

Reviewer #2: Yes

3. Has the statistical analysis been performed appropriately and rigorously? 

Reviewer #1: Yes

Reviewer #2: Yes

4. Have the authors made all data underlying the findings in their manuscript fully available?

Reviewer #1: Yes

Reviewer #2: Yes

5. Is the manuscript presented in an intelligible fashion and written in standard English?

Reviewer #1: Yes

Reviewer #2: Yes

6. Review Comments to the Author

Reviewer #1: The authors have addressed my concerns. xxxxxxxxxxxxxxxxxxxxxxxxxxxxxxxxxxxxxxxxxxxxxxxxxxxxxxxxxxx

Reviewer #2: I appreciate the efforts from authors to address the critiques from the previous review. The revised manuscript is improved and contains additional information that helps to understand the overall approach. I have no more major objections to its publication PLOS ONE. There are two minor points I want to point out here.

1. I think “relative binding energies” in Line 203 (revised manuscript) is very confusing. Considering change it to some more appropriate term like “binding enthalpies”

2. The subplots in Figure 6 look very crowded, especially the legends are overlapping with each other. I think a better way of showing the information is fitting a single line between the averaged dH to the experimental values. And move Fig 6 to SI.

7. PLOS authors have the option to publish the peer review history of their article (what does this mean?). If published, this will include your full peer review and any attached files.

Reviewer #1: No

Reviewer #2: No

---

## [Editor Report · Acceptance letter]

14 Oct 2019

PONE-D-19-23148R1 

Bridging solvent molecules mediate RNase A – ligand binding 

Dear Dr. Ivanov:

I am pleased to inform you that your manuscript has been deemed suitable for publication in PLOS ONE. Congratulations! Your manuscript is now with our production department. 

With kind regards,

on behalf of

Dr. Freddie Salsbury , Jr 

Academic Editor

PLOS ONE